# Real Life Observational Study of Treatment Success of Monoclonal Antibodies for Refractory Chronic Rhinosinusitis with Nasal Polyps

**DOI:** 10.3390/jcm12134374

**Published:** 2023-06-29

**Authors:** Kathrin Gerstacker, Manuel Christoph Ketterer, Till Fabian Jakob, Tanja Hildenbrand

**Affiliations:** Department of Oto-Rhino-Laryngology, Medical Center—University of Freiburg, Killianstr. 5, 79106 Freiburg, Germany; kathrin.gerstacker@uniklinik-freiburg.de (K.G.); manuel.christoph.ketterer@uniklinik-freiburg.de (M.C.K.); till.jakob@uniklinik-freiburg.de (T.F.J.)

**Keywords:** chronic rhinosinusitis with nasal polyps (CRSwNP), severe refractory chronic rhinosinusitis, monoclonal antibodies, real-world data, biologics

## Abstract

Chronic rhinosinusitis with nasal polyps (CRSwNP) is a mainly type-2-driven inflammatory disease that is often refractory to medical and surgical treatment and characterized by a high rate of recurrence. Monoclonal antibodies have been approved for severe refractory CRSwNP. Randomized controlled trials (RCT) have shown significant improvement in objective and subjective parameters. The results of these RCTs cannot necessarily be transferred to daily routine. The purpose of this retrospective study is to evaluate the treatment effects of biologics in patients with CRSwNP in a real-life setting. Patients treated with one of the approved biologics since July 2020 with at least 6 months follow-up were included in the study. Changes in SNOT-22 and nasal polyp score (NPS), as well as subjective change in sense of smell, number of sinus surgeries prior to and during treatment, comorbidities, discontinuation or change of monoclonal antibody and adverse events were evaluated. Thirty-three patients were included in this study. The mean SNOT-22 score and NPS improved significantly. The subjective assessment of sense of smell showed an improvement in 81.8% of patients. The monoclonal antibody had to be changed in seven patients. No severe adverse events occurred during the initiation of the treatment and follow-up. Biologics can be a treatment option in patients with severe refractory CRSwNP, showing significant improvements in quality of life, symptoms and polyp scores in randomized controlled trials and clinical routine with rare complications. Further research is needed to evaluate possible biomarkers, interdose interval prolongation and long-term safety.

## 1. Introduction

Chronic rhinosinusitis (CRS) is an inflammatory condition of the nose and paranasal sinuses, characterized by nasal obstruction, rhinorrhea, facial pain or pressure and reduction in or loss of smell. Symptoms persist for more than 12 weeks [1]. Depending on the phenotype, CRS is divided into CRS with (CRSwNP) and without (CRSsNP) nasal polyps. CRSwNP affects around 0.78–2.7% of the European population [2,3,4]. It has a significant impact on patients’ quality of life and productivity. Patients with CRSwNP often show comorbid conditions, such as asthma and aspirin-exacerbated respiratory disease (AERD).

In western countries, CRSwNP is mainly characterized by a type-2 endotype [5,6]. In brief, it is proposed that epithelial barrier dysfunction leads to the entry of antigens, irritants and pathogens. Thymic stromal lymphopoietin (TSLP) is produced and released from epithelial cells and interacts with nearby dendritic cells. The interaction of dendritic cells with naïve T cells leads to differentiation into TH 2 cells. The production of IL-25 and -33 in epithelial cells perpetuate the type-2 inflammatory response and stimulate type 2 innate lymphoid cells (ILC2). This type of inflammatory response is characterized by high levels of interleukin-4 (IL-4), interleukin-13 (IL-13), interleukin-5 (IL-5) and immunglobulin E (IgE) [1,7]. IL-4 and IL-13 induce a class switch in B cells with the synthetization of IgE. IL-5 recruits eosinophils into the tissue. These inflammatory changes induce airway remodeling with hyperplasia and metaplasia of mucus cells, fibrin mesh deposition with polyp formation in the paranasal sinuses, vascular remodeling with increasing fibrosis and hyperplasia and hypertrophy of smooth muscle cells in the lung. In addition to CRSwNP, this type-2 inflammation has been shown to play an important role in conditions such as asthma and, more recently, cystic fibrosis [1,7,8].

The first-line therapy of chronic rhinosinusitis (CRS) is medical treatment. If medical treatment fails, surgery might be indicated [1,9]. Patients with CRSwNP show high rates of refractory disease and recurrences despite adequate medical and surgical treatment. In the past, these patients had to be treated with repeated courses of systemic corticosteroids, accepting the potential adverse effects of high cumulative doses of steroids.

With the approval of monoclonal antibodies for the treatment of CRSwNP, a new treatment option is available for patients with CRSwNP refractory to medical and surgical treatment. As monoclonal antibodies target type-2 inflammatory-response components, endotyping has been integrated into clinical care pathways for patients with CRSwNP, as has been proposed in other diseases, such as asthma [7,10]. Endotypes can help to explain different outcomes in patients with similar phenotypes. They can also help with choosing between treatment options and estimating prognosis. Strong indicators for a type-2 endotype in CRSwNP are late-onset asthma, tissue eosinophilia, blood eosinophils > 300 cells/μL, total serum IgE > 150 kU/L and the presence of S. aureus enterotoxin-specific IgE (SE-IgE). The combination of late-onset asthma and blood eosinophils > 300 cells/μL has a high specificity and sensitivity for a type-2 endotype in CRSwNP [7].

Three monoclonal antibodies have been approved as an add-on treatment to topical corticosteroids in patients with severe refractory CRS in Germany: Dupilumab, Omalizumab and Mepolizumab.

Dupilumab is a human monoclonal antibody that targets IL4Rα, inhibiting IL-4 and IL-13. Omalizumab and Mepolizumab are humanized monoclonal antibodies that block IgE and IL-5, respectively.

The randomized controlled trials that led to their approval were performed for all three monoclonal antibodies [11,12,13]. These trials were LIBERTY NP SINUS-24/-52 for Dupilumab, POLYP 1 & 2 for Omalizumab and SYNAPSE for Mepolizumab. They were able to show that there is improvement in Sino-Nasal Outcome Test-22 (SNOT-22), nasal polyp score (NPS), nasal congestion score or nasal obstruction VAS score, symptom scores, sense of smell, CT scores and proportion of patients requiring surgery or systemic corticosteroids.

Patients in these trials were highly selected according to inclusion and exclusion criteria and did not necessarily match those patients treated in daily routine. Therefore, the aim of this study is to evaluate the effects of monoclonal antibody treatments in patients with severe recalcitrant CRSwNP in an otolaryngology department of a tertiary medical center. We hypothesize that patients with severe recalcitrant CRSwNP treated with monoclonal antibodies will show improvement of subjective and objective outcome parameters in a real-life setting, as has been shown in the randomized controlled trials.

## 2. Methods

### 2.1. Study Design

The present study is a monocentric, retrospective study. The study was conducted at the Department of Oto-Rhino-Laryngology of the University of Freiburg Medical Center. All patients treated with one of the approved biologics for recalcitrant chronic rhinosinusitis with nasal polyps since 2020 and who completed at least 6 months of follow-up were included in the study.

### 2.2. Study Population

CRSwNP was diagnosed according to the European Position Paper on Rhinosinusitis and Nasal Polyps (EPOS) criteria [1]. The indication for monoclonal antibody treatment was provided by an experienced rhinologist according to the approval criteria and international and national guidelines [1,14,15]. All patients showed severe recalcitrant CRSwNP despite appropriate medical (systemic corticosteroids and intranasal corticosteroids and nasal saline irrigation) and surgical treatment with persistent symptoms of chronic rhinosinusitis and bilateral nasal polyps with a minimum NPS of 2 on each side. As all biologics are approved as an add-on therapy to topical nasal steroids, all patients were on long-term intranasal corticosteroids (INCS). All biologics are approved for treatment of patients ≥ 18 years of age; therefore, all patients were older than 18 years of age. Dupilumab 300 mg was administered subcutaneously every 4 weeks, Mepolizumab 100 mg every 4 weeks and Omalizumab depending on total IgE and bodyweight every 4 weeks. The first 2 to 4 injections were provided at the outpatient department of the department of otorhinolaryngology to educate patients and to ensure that there were no early adverse events. Patients were scheduled for follow-up appointments after 3 months, 6 months and 12 months. Response to treatment was evaluated according to EPOS guidelines [1]. Response was evaluated after 6 months, if no adverse events occurred prior to this timepoint. If there was no response, the patient was counseled and change in monoclonal antibody or, if indicated, revision surgery was offered. If there was a partial response, the continuation of treatment was discussed with the patient. This is in accordance with national and international treatment recommendations [16,17,18].

### 2.3. Ethical Considerations

The study was conducted according to national regulations and the 1964 declaration of Helsinki and its later amendments as well as the ‘Note for Guidance on Good Clinical Practise’ (GCP) from 17 January 1997. The study was approved by the ethics committee of the University of Freiburg (22-1497-S1-retro). The participants were provided an information sheet about the study and provided written informed consent. They were also informed about their rights according to current data protection regulations and provided written consent for the collection, analysis and storage of their personal data.

### 2.4. Data Collection

The analysis was based on statistical data from the Department of Oto-Rhino-Laryngology, the patients’ medical records and digitized examination results.

### 2.5. Primary Outcome Parameters

The primary objectives of the study were change in quality of life, evaluated by SNOT-22 (©2006, Washington University, St. Louis, MO, USA), and change in nasal polyp score (NPS) as differences from baseline to 4 weeks, 3, 6 and 12 months of treatment. The SNOT-22 is a disease-specific patient reported outcome measure. It consists of 22 items that can be categorized into the rhinologic symptoms domain, extra-nasal rhinologic symptoms domain, ear/facial symptoms domain, psychological dysfunction domain and the sleep dysfunction domain [19]. It is used in most studies on monoclonal antibody treatment and is well established in CRS research. It has been translated and validated in different languages. The German-validated version was used for this study [20]. The minimal clinically important difference (MCID) was found to be 8.9 points [21]. The NPS was evaluated by nasal endoscopy for both sides. The scores are defined as score 0: no polyps on nasal endoscopy; score 1: small polyps in the middle meatus not reaching below the inferior border of the middle turbinate; score 2: polyps reaching below the lower border of the middle turbinate; score 3: large polyps reaching the lower border of the inferior turbinate or polyps medial the middle turbinate; and score 4: large polyps causing complete obstruction of the inferior nasal cavity [22,23].

### 2.6. Secondary Outcome Parameters

Age, gender, clinical history, number of sinus surgeries prior to and during treatment, previous systemic corticosteroid treatment, comorbidities (asthma, intolerance to non-steroidal anti-inflammatory drugs (NSAID) and AERD), eosinophilic count and tissue eosinophilia were recorded, as these can be possible confounders influencing the outcome. In addition, the subjective improvement of sense of smell, discontinuation or change in monoclonal antibody and adverse events were evaluated as secondary outcomes.

### 2.7. Statistical Analysis

Statistical analysis was performed with Excel (Microsoft Corporation, Redmond, WA, USA) and IBM SPSS Statistics (IBM Corporation, Released 2015. IBM SPSS Statistics for Windows, Version 24.0, Armonk, NY, USA) and the level of significance was set at *p* < 0.05. The results were calculated with median, standard deviation and minimum/maximum. We calculated the comparison analysis using a Pearson-Chi-Quadrat test and to compare means and the statistical significance, we uses a *t*-test. We compared the primary outcome parameters for patients with and without asthma using a comparing median *t*-test. Excel (Microsoft Corporation) was used to compile the figures.

## 3. Results

Forty-two patients were started on a monoclonal antibody treatment between July 2020 and December 2022. Thirty-three patients (m = 19, f = 14, mean age 47.3 years, SD +/−12.97)) completed at least 6 months of follow-up and were included in this study. Nine patients with a follow-up period of less than 6 months were excluded from the study. Six months of follow-up data were available for 33 patients and twelve months of follow-up data for 19 patients.

The epidemiologic data and relevant comorbidities are summarized in Table 1.

A total of 72.7% of patients were asthmatic, 51.5% suffered from intolerance to NSAID and 45.5% showed the typical triad of aspirin-exacerbated respiratory disease (AERD). All patients reported at least one course of systemic steroids and one previous sinus surgery (min 1, max 15, M = 2.9, SD +/−2.5). Histological results were available for 22 patients. All but one patient showed tissue eosinophilia in surgical specimens. Eleven patients had surgery elsewhere and histological results were not available. Blood eosinophils were available for 25 patients. Sixteen of these patients showed blood eosinophils of >300 cells per microliter.

Mean SNOT-22 score prior to treatment was 59.8 (SD +/−15.1). This improved to 41.5 (SD +/−19.8) after 4 weeks, 31.1 (SD +/−18.3) after 3 months, 27 (SD +/−17.1) after 6 months and 15.6 (SD +/−11.3) after 12 months. Mean differences from baseline were −18.33 after 4 weeks, −28.72 after 3 months, −32.82 after 6 months and −44.23 after 12 months (Figure 1). These differences were significant from 4 weeks to 12 months of follow-up (*p* > 0.001). There was no statistically significant difference between patients with asthma and without asthma at baseline and up to 6 months of follow-up.

Mean pre-treatment NPS was 5.03 (SD +/−1.2). NPS decreased to 3.79 (SD +/−1.9) after 4 weeks, 2.58 (SD +/−2.1) after 3 months, 2.19 (SD +/−2.2) after 6 months and 1.05 (SD +/−1.6) after 12 months. Mean differences from baseline were −1.24 after 4 weeks, −2.45 after 3 months, −2.84 after 6 months and −3.98 after 12 months (Figure 2). These differences were significant from 4 weeks to 12 months (*p* > 0.001) of follow-up. There was no statistically significant difference between patients with asthma and without asthma at baseline and up to 6 months’ follow-up.

The subjective assessment of the sense of smell showed an improvement in 81.8% of patients. Seventeen patients still complained of hyposmia and two of anosmia.

Twenty-eight patients were initially treated with Dupilumab, three with Omalizumab and two with Mepolizumab. The monoclonal antibody had to be changed in seven patients. This was due to the lack of subjective and objective improvement in five patients and complications in two patients. Three patients were changed from Omalizumab to Dupilumab, two from Dupilumab to Mepolizumab and two from Mepolizumab to Omalizumab. In one patient, the biologic treatment had to be changed twice. This patient showed persistent high eosinophilic counts of up to 4000/μL during the initiation of Dupilumab treatment and was changed to Mepolizumab. Due to a lack of improvement, he was changed again to Omalizumab and is at present showing a good control of his disease. Another patient complained of intense abdominal pain and headaches after the first injections of Dupilumab and the treatment was changed to Mepolizumab.

No severe adverse events occurred during the initiation of the treatment and the follow-up period. One patient complained of abdominal pain, one of dizziness and two of headaches during the initiation of the treatment, which subsided during further treatment, except in one patient in whom the monoclonal antibody had to be changed. Four patients showed a marked eosinophilia of >1500/μL during the initiation of the Dupilumab treatment. None of these patients showed clinical signs of hypereosinophilic syndrome. Eosinophilic counts decreased during further treatment. One patient had to be treated with antibiotics for acute exacerbation of CRS during the study period. One patient discontinued treatment after 10 months on Dupilumab due to Herpes zoster, recommended by the treating neurologist. One patient had surgery for the removal of single polyps in the anterior ethmoid bilaterally. No other patient had surgery during the study period. The secondary outcome parameters are specified in Table 2.

## 4. Discussion

CRSwNP is often difficult to treat. Patients show a high burden of disease despite appropriate medical and surgical treatment. With the approval of three monoclonal antibodies for the treatment of CRSwNP, there is an additional treatment option for these patients. Our patients received all three of the available biologics, although due to the date of approval, less patients were treated with Omalizumab and Mepolizumab, compared to Dupilumab. Due to the much smaller number of patients treated with Mepolizumab and Omalizumab, the comparison of treatment effects of the different monoclonal antibodies was not feasible.

Comparing our patients to the study cohorts of the randomized controlled trials, our patients showed higher rates of comorbid asthma and AERD and a higher rate of previous surgeries. This might be due to the fact that the included patients were the first cohort that was started on biologics in our department and therefore only the most severely affected patients were selected for treatment. Patients with comorbid asthma and AERD are known to show higher recurrence rates and higher rates of refractory disease. All of our patients had at least one previous surgery. Although this is not a prerequisite for monoclonal antibody treatment, surgery is currently still considered as a first-line treatment in patients failing medical treatment [1,9]. It has also been shown that functional endoscopic sinus surgery (FESS) is currently more cost-effective than treatment with biologics [24].

Mean SNOT-22 and polyp scores at baseline in our study were comparable to the randomized controlled trials [11,12,13].

The randomized controlled trials showed significant improvement in NPS and quality of life. Although direct comparison is limited due to the smaller sample size in our study, we could also show significant improvement in both primary outcome parameters as early as four weeks after the start of treatment. The mean improvement in SNOT-22 score exceeded the MCID of 8.9 as early as four weeks after the initiation of treatment, i.e., that the improvement is noticeable and meaningful for the patients. SNOT-22 and NPS improved throughout the follow-up period with further improvement even after 6 months. The improvements in NPS and SNOT-22 were more pronounced than in the randomized controlled trials. This might again be due to the selection of the most severely affected patients with strong signs of type-2 inflammation in most patients. There was no significant difference in SNOT-22 and NPS between patients with and without asthma at baseline and up to 6 months of follow-up. This is in line with other studies [11,25,26]. In contrast, the real-life study by Haxel et al. showed that asthma had a significant effect on the response to treatment [27].

Only one patient showed no improvement in nasal polyp score, although the treatment was changed from Dupilumab to Mepolizumab after 6 months. Nevertheless, this patient reported a considerable decrease in SNOT-22 score from 50 to 29 and an improvement in the sense of smell from anosmia to hyposmia. The patient was offered surgery with continued biologic treatment or a change to Omalizumab, but was reluctant due to the relatively low symptom burden and clear improvement in quality of life. His NPS at baseline was 3 bilaterally. He showed no tissue eosinophilia in biopsies taken at his first surgery, a baseline eosinophilic count of 200/μL, total IgE of 193 IU/mL and he is not known to be asthmatic. He had previously been tested for cystic fibrosis and eosinophilic granulomatosis with polyangiitis (EGPA), and all tests were negative. Taking this into account, his CRSwNP might be caused by a type-1 inflammation that might explain the lack of improvement in NPS. This highlights the importance of careful patient selection and the need for biomarkers.

The sense of smell improved in 81.8% of our patients. Although we were not able to include psychophysical testing, this is still impressive, as the ability to smell has a high impact on patients’ quality of life. However, the sense of smell did not improve in roughly one fifth of our patients, even in those with no evidence of residual polyps or edema in the olfactory cleft and seventeen patients still stated to be hyposmic. This could be due to residual inflammation within the olfactory mucosa and irreversible damage due to long standing inflammation or repeated surgery.

The monoclonal antibody had to be changed in seven patients, which was due to a lack of subjective and objective improvement in five patients. To date, there are no biomarkers to determine which biologic to choose for the individual patient to predict response to treatment. In addition, there are no head-to-head trials comparing the efficacy of the different biologics. As long as this is the case, the choice of the monoclonal antibody depends on comorbidities, such as eosinophilic granulomatosis with polyangiitis (EGPA) and physicians’ and patients’ preferences.

No severe adverse events occurred during the initiation of the treatment or the follow-up period. All adverse events that occurred were mild and subsided without further treatment, so safety was comparable to other published studies. Adverse events reported in these studies were mainly headaches, injection site erythema, arthralgia, nasopharyngitis, conjunctivitis, epistaxis, nasal polyps, asthma and acute sinusitis [11,12,13,27,28].

The results of placebo-controlled randomized studies cannot necessarily be transferred to the reality in patient care as there are specified inclusion and exclusion criteria that do not match a real clinical setting. Furthermore, randomized PCT effects are expected from participation in the study due to the regular consequent use of medication and the placebo effect. For this reason, it is important to evaluate treatment success in a real-life setting. Some real-life studies have been published to date [27,28,29,30,31,32,33,34,35]. Most of these studies are retrospective studies except for the studies by van der Lans and Haxel et al. Van der Lans et al. prospectively enrolled 131 patients treated with Dupilumab and Haxel et al. 70 patients treated with either Dupilumab or Omalizumab. Most of the real-world studies include a very small number of patients and are quite inhomogeneous with regard to outcome measures. In some of these studies, the indication for biologic treatment was severe asthma or EGPA with comorbid CRSwNP [15,30]. Nevertheless, all the mentioned studies show a significant improvement in the investigated parameters, such as NPS and SNOT-22. The response rates were reported to range from 50% up to 100%, although response was defined differently in the individual studies.

Compared to the study of van der Lans et al. and Haxel et al., our patients showed comparable NPS and SNOT-22 score at baseline and after 12 months (only in van der Lans et al.). At 6 months of follow-up, our patients’ NPS and SNOT-22 score were slightly higher compared to those of van der Lans et al. and Haxel et al. [27,33].

One limitation of our study is the lack of psychophysical testing of the sense of smell, which was due to COVID-19 regulations at our hospital. Sniffin’ sticks tests were not consistently performed and the limited data could not be used for analysis. More precise testing of olfactory ability is desirable, e.g., with Sniffin’ sticks tests, which we were able to resume to perform in all patients. We at present also use a sense of smell VAS for further evaluation. Many studies use the University of Pennsylvania Smell Identification Test (UPSIT), but it is not commonly used in Germany. It has been shown that olfaction measured by Sniffin’ sticks identification test improves in patients with CRSwNP with monoclonal antibody treatment [27,28]. Furthermore, selection bias may have occurred due to the setting of the study at a university hospital. Further bias could be introduced by the fact that the study cohort comprised the first patients treated with monoclonal antibodies at our department; so, the selected patients could represent a group of patients with most severe, difficult-to-treat disease. This could overestimate the treatment effect of monoclonal antibodies. A prospective multicenter study with clearly defined inclusion/exclusion criteria would limit these biases. Only one patient had sinus surgery during the follow-up period, but the observation period of 6 to 12 months is too short to draw a definite conclusion of the extent to which the need for revision surgery is reduced. However, it can be assumed that the reduction in symptoms and improvement in polyp score and quality of life will lead to a reduction in the need for revision surgery. The generalizability of our results is limited by the small number of patients included in the study. A larger multicenter study is desirable to provide more robust evidence.

## 5. Conclusions

The real-life data of our retrospective study seem to confirm that monoclonal antibodies can be effective in the treatment of recalcitrant CRSwNP. The patients showed an improvement in symptoms, such as sense of smell, quality of life and endoscopic NPS. The rate of side effects seems to be low. The indication should be well reviewed and provided according to guidelines and in the awareness of high annual costs, on the one hand, and the possibility of avoiding revision surgery, medications and hospitalization, on the other hand. As long as there are no biomarkers, the choice of monoclonal antibody must be based on individual factors, such as comorbidities and physicians’ and patients’ preferences.

## Figures and Tables

**Figure 1 jcm-12-04374-f001:**
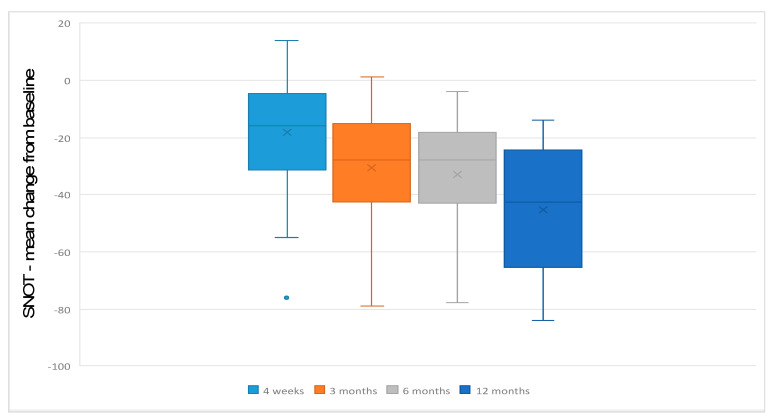
Boxplot displaying the mean change from baseline in SNOT-22 total score 4 weeks, 3, 6 and 12 months after the initiation of the treatment.

**Figure 2 jcm-12-04374-f002:**
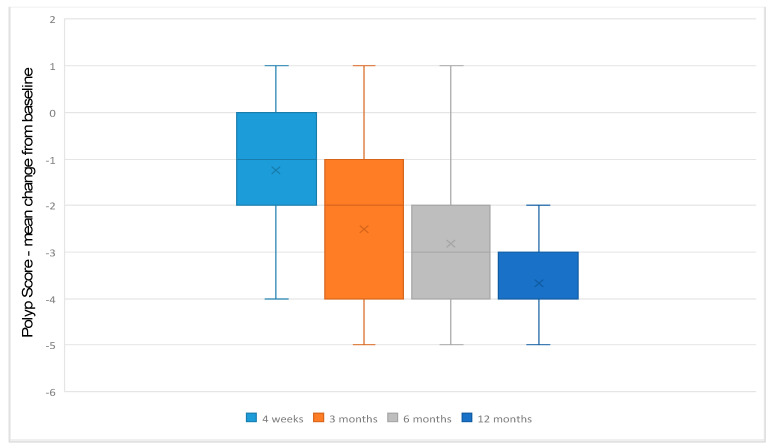
Boxplot displaying the mean change from baseline in NPS 4 weeks, 3, 6 and 12 months after the initiation of the treatment.

**Table 1 jcm-12-04374-t001:** Epidemiologic data of the study population.

Epidemiologic Data	
**Gender** Male Female	19 14
**Age** Range Mean	22–70 years 47.3 (SD +/−12.97)
**Comorbidities (% of patients)** Asthma Aspirin-exacerbated respiratory disease (AERD)	72.7% 45.5%
**Eosinophilia** Tissue eosinophilia Blood eosinophils > 300/μL	21/22 16/25
**Previous corticosteroid treatment** Topical steroids Systemic steroids > 1 course	100% 100%
**Previous surgeries** Range Mean ≥2 surgeries (% of patients)	1–15 2.9 (SD +/−2.5) 75.8%

**Table 2 jcm-12-04374-t002:** Secondary outcome parameters of the study population.

Secondary Outcome Parameters	
**Surgery during treatment**	1/33
**Subjective improvement of sense of smell**	81.8%
**Discontinuation of treatment**	1/33
**Change in monoclonal antibody**	7/33
**Adverse Events** Abdominal pain Headaches Dizziness Eosinophilia Acute exacerbation of CRS	8/33 1/33 2/33 1/33 4/33 1/33

## Data Availability

No new data were created or analyzed in this study. Data sharing is not applicable to this article.

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
