# Peer review of "Real Life Observational Study of Treatment Success of Monoclonal Antibodies for Refractory Chronic Rhinosinusitis with Nasal Polyps"

_jcm, 2023, doi:10.3390/jcm12134374_

Round 1

Reviewer 1 Report

Dear Editor

many thanks for asking me to review this original article aiming to evaluate retrospectively the treatment effects of biologics in patients with CRSwNP in a real life setting.

Looking at the introduction section: the rationale of the efficacy of biologic drugs in CRSwNP should include a brief focus on the role of Type2 inflammation on the onset of this disease. Please, add more information also referring to: Type 2 inflammation in cystic fibrosis: New insights. Pediatr Allergy Immunol. 2022;33 Suppl 27(Suppl 27):15-17

I am surprising by the high incidence of asthma and AERD detected in the enrolled population. Please, could you try to provide an explanation? 

Regarding the safety, please provide more information. The sentence "All adverse events that occurred were mild and subsided without further treat ment" is not sufficient to achieve definitive conclusions.

Author Response

Dear reviewer,

thank you very much for your comments to improve our manuscript. We have made changes according to your suggestions. We would like to answer to your comments as follows:

We added more details of the type-2 inflammatory response in the pathophysiology of CRSwNP and in other diseases such as asthma and CF in the introduction and added the proposed article to the references.

We think that the high rate of comorbid asthma and AERD is most likely due to the fact that the patients presented in this study were the first patients treated with biologics in our department, so only the most severely affected patients with severe recurrent disease have been selected for treatment. Asthma and AERD are known to lead to a higher rate of recurrence and refractory courses, so patients with severe refractory CRSwNP often show these comorbidities. We highlighted the relevant paragraph in the discussion and complemented it.

Adverse events of our study cohort are depicted in table 2 and stated in the results section. We added the most important adverse events that were reported in most other studies to the discussion.

Thank you for taking the time to review our manuscript.

Reviewer 2 Report

This study provides valuable real-world evidence on the effectiveness of monoclonal antibodies for recalcitrant CRSwNP. Randomized controlled trials, while the gold standard for evaluating treatment effects have limited generalizability to real-world practice due to strict inclusion/exclusion criteria and placebo effects. Real-world data studies complement RCTs by demonstrating how interventions perform in typical heterogenous patient populations and practice settings. This study found that in a real-world tertiary care setting, monoclonal antibodies improved both symptoms (SNOT-22) and objective outcomes (NPS) over a follow-up period of 6-12 months in patients with severe CRSwNP refractory to medical and surgical interventions. The study follow-up period allowed assessment of both short-term (6 months) and longer-term (12 months) outcomes, with continued improvement noted up to 12 months.

The study also utilized both subjective and objective outcome measures providing a comprehensive assessment of treatment effects. The SNOT-22 questionnaire is a validated patient-reported outcome measure that provides an assessment of the health-related quality of life specific to the sinonasal disease. The NPS provides an objective assessment of nasal polyp size and was evaluated by an experienced rhinologist, limiting potential bias that can occur with patient self-report. The use of both symptomatic and objective measures is a strength, as certain interventions may have greater effects on either objective or subjective outcomes. This study found significant improvements in both SNOT-22 scores and NPS, suggesting a meaningful treatment benefit, both symptomatically and objectively.

Suggestions for improvement:

  • Introduction
  • - With the approval of monoclonal antibodies for the treatment of CRSwNP, a new treatment option is available for patients with CRSwNP refractory to medical and surgical treatment. As monoclonal antibodies target type-2 inflammatory response components, endotyping has been integrated into clinical care pathways for patients with CRSwNP.. please discuss and cite doi:10.1016/j.jaci.2015.12.1322
    and doi:10.1016/j.jaci.2019.01.003

  • - while preoperative nasal surgery is commonly performed, it is not always required prior to initiating biologic treatment for CRSwNP. The necessity for surgery depends on individual patient and disease factors. Either with or without surgery, biologics show benefit in appropriately selected patients with severe CRS refractory to other treatments., please discuss and cite doi:10.1007/s00405-021-06724-6
  • The study had a small sample size (n=33) limiting the power and generalizability of the results. A larger multi-center study would provide more robust evidence.
  • adapt to equator guidelines to improve the structure
  • There was potential selection bias as the study was conducted at a tertiary care center and included the first patients started on monoclonal antibodies. A prospective study with clearly defined inclusion/exclusion criteria would limit selection bias.
  • There was lack of control group for comparison. A randomized controlled trial comparing monoclonal antibodies to placebo or surgery would provide higher quality evidence.
  • A randomized controlled trial that included a placebo control group for evaluating biologic treatment of CRSwNP is the SYNAPSE study published by Bachert et al. in 2020. This 24-week double-blind trial compared dupilumab, an interleukin-4 receptor alpha inhibitor, to placebo in 60 patients with severe CRSwNP who required systemic corticosteroid therapy. please discuss and cite doi:10.1016/j.otohns.2020.01.012
  • Sense of smell was assessed subjectively only. Inclusion of objective testing of olfaction such as sniffin sticks would strengthen the evidence related to effects on sense of smell.
  • A randomized controlled trial that used the Sniffin' Sticks test to evaluate olfaction in patients with CRSwNP treated with biologics is the OSTRICH trial published by Vaidyanathan et al. in 2019. This 24-week trial compared dupilumab to placebo in 60 patients with CRSwNP. along with other measures, the OSTRICH study used the Sniffin' Sticks Extended Test to assess any changes in objective olfactory function. please discuss and cite https://doi.org/10.1016/j.jaci.2019.04.043
  • Stratifying outcomes by specific monoclonal antibody received would allow comparison of treatment effects between agents. As more patients receive Omalizumab and Mepolizumab, analyzing outcomes by specific drug will be informative.
  • The SNOT-22, while validated and widely used, contains 22 items which can be time consuming for patients to complete and for clinicians to score and interpret. To address this, a shortened version of the SNOT-22 was developed, the SNOT-16 and SNOT-8.
    The SNOT-8 has been shown to have good internal consistency, test-retest reliability, and correlation with the original SNOT-22. The SNOT-8 may reduce patient burden and allow for greater feasibility of use, without loss of measurement sensitivity or validity.
    In conclusion, the SNOT-8 provides a shortened yet valid version of the original, please discuss and cite 10.1007/s00405-023-07855-8 and doi:10.1002/alr.23002
  • Including data on healthcare utilization such as hospitalizations, surgeries and corticosteroid use would help determine the economic impact of treatment.
  • Discussing limitations of the study including lack of biomarker data to determine optimal treatment selection would provide important context for readers.

Overall this study provides useful real-world data on the effectiveness of monoclonal antibodies for a difficult to treat condition. Conducting a larger, prospective study with a control group and objective outcome measures would build on these results and provide higher quality evidence to guide the management of recalcitrant CRSwNP.

Author Response

Dear reviewer,

thank you very much for your comments to improve our manuscript. We have made changes according to your suggestions. We would like to answer to your comments as follows:

  • - With the approval of monoclonal antibodies for the treatment of CRSwNP, a new treatment option is available for patients with CRSwNP refractory to medical and surgical treatment. As monoclonal antibodies target type-2 inflammatory response components, endotyping has been integrated into clinical care pathways for patients with CRSwNP.. please discuss and cite doi:10.1016/j.jaci.2015.12.1322 and doi:10.1016/j.jaci.2019.01.003

We’re not sure if the studies retrieved by our doi search in Pub Med and on the doi Foundation homepage are the ones you recommended to discuss and cite as they discuss different topics not clearly related to our study.

doi: 10.1016/j.jaci.2015.12.1322: Michael R. Perkin, Kirsty Logan, Tom Marrs, Suzana Radulovic, Joanna Craven, Carsten Flohr, Gideon Lack, and EAT Study Team. Enquiring About Tolerance (EAT) study: Feasibility of an early allergenic food introduction regimen. J Allergy Clin Immunol. 2016 May; 137(5): 1477–1486.e8.

doi: 10.1016/j.jaci.2019.01.003.:Wendy F Davidson, Donald Y M Leung, Lisa A Bec k, Cecilia M Berin, Mark Boguniewicz, William W Busse, Talal A Chatila, Raif S Geha, James E Gern, Emma Guttman-Yassky, Alan D Irvine, Brian S Kim, Heidi H Kong, Gideon Lack   , Kari C Nadeau, Julie Schwaninger, Angela Simpson, Eric L Simpson, Jonathan M Spergel, Alkis Togias, Ulrich Wahn, Robert A Wood, Judith A Woodfolk, Steven F Ziegler, Marshall Plaut. Report from the National Institute of Allergy and Infectious Diseases workshop on "Atopic dermatitis and the atopic march: Mechanisms and interventions". J Allergy Clin Immunol. 2019 Mar;143(3):894-913.

The Report from the National Institute of Allergy and Infectious Diseases workshop on "Atopic dermatitis and the atopic march: Mechanisms and interventions" discusses recommendation of studies on endotyping in atopic dermatitis to predict progression. Therefore clinical care pathways don’t exist as yet. Nevertheless, we added a paragraph on endotyping in CRSwNP.

  • - while preoperative nasal surgery is commonly performed, it is not always required prior to initiating biologic treatment for CRSwNP. The necessity for surgery depends on individual patient and disease factors. Either with or without surgery, biologics show benefit in appropriately selected patients with severe CRS refractory to other treatments., please discuss and cite doi:10.1007/s00405-021-06724-6

Again, we’re not sure if this is the right article as it deals with surgery for contact point headaches.

Antonino Maniaci  1 , Federico Merlino  2 , Salvatore Cocuzza  2 , Giannicola Iannella  3   4 , Claudio Vicini  4 , Giovanni Cammaroto  4 , Jérome R Lechien  5   6   7   8 , Christian Calvo-Henriquez  9 , Ignazio La Mantia  2. Endoscopic surgical treatment for rhinogenic contact point headache: systematic review and meta-analysis. Eur Arch Otorhinolaryngol. 2021 Jun;278(6):1743-1753.

Your right, surgery is not required prior to biologics treatment. But it is still recommended as first line treatment for patients failing medical treatment by many international guidelines, if there are no contraindications. As biologics are expensive and need to be given long-term, currently FESS is more cost effective (Scangas GA, Wu AW, Ting JY, Metson R, Walgama E, Shrime MG, et al. Cost utility analysis of dupilumab versus endoscopic sinus surgery for chronic rhinosinusitis with nasal polyps. Laryngoscope 2021;131:E26-33.). We added this to our discussion.

  • The study had a small sample size (n=33) limiting the power and generalizability of the results. A larger multi-center study would provide more robust evidence.
  • There was potential selection bias as the study was conducted at a tertiary care center and included the first patients started on monoclonal antibodies. A prospective study with clearly defined inclusion/exclusion criteria would limit selection bias.

We agree with your comments. We discuss this in the paragraph on limitations of the study.

  • adapt to equator guidelines to improve the structure

As the editor suggested this as well, we used the STROBE criteria to revise the manuscript.

  • There was lack of control group for comparison. A randomized controlled trial comparing monoclonal antibodies to placebo or surgery would provide higher quality evidence.

You’re right. There are high quality randomized controlled trial comparing monoclonal antibodies to placebo, which is the gold standard. But this is not the purpose of real-world evidence, as these PCT do not reflect treatment in every day practice.

  • A randomized controlled trial that included a placebo control group for evaluating biologic treatment of CRSwNP is the SYNAPSE study published by Bachert et al. in 2020. This 24-week double-blind trial compared dupilumab, an interleukin-4 receptor alpha inhibitor, to placebo in 60 patients with severe CRSwNP who required systemic corticosteroid therapy. please discuss and cite doi:10.1016/j.otohns.2020.01.012 doi:10.1016/S2213-2600(21)00097-7.

We think you meant the SINUS-24 and -52 study on Dupilumab. We’ve cited it several times in the manuscript.

  • Sense of smell was assessed subjectively only. Inclusion of objective testing of olfaction such as sniffin sticks would strengthen the evidence related to effects on sense of smell.

You’re absolutely right. Unfortunately we started treatment during the COVID pandemic. Due to official regulations by the hospital we were not allowed to perform sniffin’ sticks tests for many months. Due to this, sniffin’ sticks test results were not available for some of the patients and the data volume was not suitable for analysis. We included this in the limitation section.

  • A randomized controlled trial that used the Sniffin' Sticks test to evaluate olfaction in patients with CRSwNP treated with biologics is the OSTRICH trial published by Vaidyanathan et al. in 2019. This 24-week trial compared dupilumab to placebo in 60 patients with CRSwNP. along with other measures, the OSTRICH study used the Sniffin' Sticks Extended Test to assess any changes in objective olfactory function. please discuss and cite https://doi.org/10.1016/j.jaci.2019.04.043

Unfortunately again, we couldn’t find the proposed study, but we included two real-world studies using the Sniffin’ sticks test for evaluating olfaction.

  • Stratifying outcomes by specific monoclonal antibody received would allow comparison of treatment effects between agents. As more patients receive Omalizumab and Mepolizumab, analyzing outcomes by specific drug will be informative.

Again, we fully agree. It would be of great help to have these treatment comparisons of different monoclonal antibodies to guide selection of a biologic for a specific patient. As randomized trials for head to head comparisons of different monoclonal antibodies are currently performed, the results will be much anticipated.

  • The SNOT-22, while validated and widely used, contains 22 items which can be time consuming for patients to complete and for clinicians to score and interpret. To address this, a shortened version of the SNOT-22 was developed, the SNOT-16 and SNOT-8. The SNOT-8 has been shown to have good internal consistency, test-retest reliability, and correlation with the original SNOT-22. The SNOT-8 may reduce patient burden and allow for greater feasibility of use, without loss of measurement sensitivity or validity. In conclusion, the SNOT-8 provides a shortened yet valid version of the original, please discuss and cite 10.1007/s00405-023-07855-8 and doi:10.1002/alr.23002

Unfortunately the SNOT-8 is not available in a German validated version. Furthermore the SNOT-22 is the most widely used specific health related quality of life questionnaire in CRS research, especially regarding monoclonal antibody treatment. Therefore we think that it should be used to allow for comparison of outcomes of different studies.

  • Including data on healthcare utilization such as hospitalizations, surgeries and corticosteroid use would help determine the economic impact of treatment.

This is true. Unfortunately it is difficult to include this now, as the ethics committee vote doesn’t include this.

  • Discussing limitations of the study including lack of biomarker data to determine optimal treatment selection would provide important context for readers.

We include this in the discussion and highlighted it in the revision to improve traceability.

Thank you again for taking the time to review our manuscript